# "The system is a bit broken…" a qualitative exploration of barriers in the pathway for diagnosing Developmental Coordination Disorder

Lucy H. Eddy [1]*, Nat K. Merrick[2], Cara E. Staniforth[3], Jade L. Jukes[4], Liam J. B. Hill[5], Mark Mon-Williams[6,7,8], Farid Bardid[9,10], Rebecca Murray[4]

**1** School of Psychology, Northumbria University, Newcastle, Tyne and Wear, United Kingdom, **2** Faculty of Health and Social Care, University of Bradford, Bradford, West Yorkshire, United Kingdom, **3** Institute of Health and Social Care Research, University of Bradford, Bradford, West Yorkshire, United Kingdom, **4** Department of Psychology, University of Bradford, Bradford, West Yorkshire, United Kingdom, **5** Moray House School of Education and Sport, University of Edinburgh, Edinburgh, Midlothian, United Kingdom, **6** School of Psychology, University of Leeds, Leeds, West Yorkshire, United Kingdom, **7** Born In Bradford's Centre for Applied Education Research, Bradford, West Yorkshire, United Kingdom, **8** National Centre for Optics, Vision and Eye Care, University of South-Eastern Norway, Notodden, Norway, **9** Strathclyde Institute of Education, University of Strathclyde, Glasgow, Lanarkshire, United Kingdom, **10** Department of Psychological Sciences and Health, University of Strathclyde, Glasgow, UK

\* lucy.eddy@northumbria.ac.uk

## Abstract

### Background

Approximately 5% of children are affected by a neurodevelopmental disorder of their sensorimotor skills. DSM-V and ICD-10, the two most widely used diagnostic systems, define this diagnostically as 'Developmental Coordination Disorder' (DCD) or 'Specific Developmental Disorder of Motor Function' (SDDMF), respectively. A diagnosis of DCD has been found to have a detrimental impact on a range of outcomes (e.g., health and education). It is therefore crucial that these children receive timely intervention. This is reliant, however, on effective assessment and support pathways. Research has shown there is great parental dissatisfaction, but there has been limited research exploring a clinical and education perspective. This study therefore aimed to understand barriers and facilitators for clinical and education practitioners in the pathway in a diverse district in the UK (Bradford).

### Methods

Semi-structured interviews were completed with stakeholders across the pathway to identify barriers and facilitators to assessing, diagnosing, and supporting children with sensorimotor skill difficulties. Theoretical thematic analysis aligned to the Capability, Opportunity, Motivation model of Behaviour change (COM-B) was used to analyse the qualitative data.

**Data availability statement:** Due to the small number of professionals in some of the roles interviewed, it is not possible to provide transcripts as these individuals would be easily identified. Upon reasonable request, authors will share transcripts which remove any reference to participants' job role. To access the data in an anonymised and redacted format, individuals can contact the ethics committee (ethics@bradford.ac.uk).

**Funding:** The work of the lead author (L.H. Eddy) and co-author (C.E. Staniforth) was supported by a grant from the Waterloo Foundation (ref: 27665413). M. Mon-Williams was supported by a Fellowship from the Alan Turing Institute. The work was conducted within infrastructure provided by the Centre for Applied Education Research (funded by the Department for Education through the Bradford Opportunity Area) and ActEarly: a City Collaboratory approach to early promotion of good health and wellbeing funded by the Medical Research Council (grant reference MR/S037527/). M. Mon-Williams' involvement was supported by the National Institute for Health Research Yorkshire and Humber ARC (reference: NIHR20016). The views expressed in this publication are those of the author(s) and not necessarily those of the National Institute for Health Research or the Departments of Health and Social Care or Education. The funders had no role in study design, data collection and analysis, decision to publish, or preparation of the manuscript.

**Competing interests:** The authors have declared that no competing interests exist.

## Results

Interviews revealed many barriers in the DCD pathway related to *capability* (confusing terminology, inconsistent knowledge, inappropriate referrals), *opportunity* (resource constraints, DCD being considered low priority, and disconnected services), and *motivation* (overlapping job roles, a desire to consider those with difficulties not eligible for a diagnosis). No facilitators were consistently identified across interviews.

## Conclusion

Families face multiple barriers to obtaining a diagnosis for their child through existing clinical pathways for assessment and support. These findings are unlikely to be unique to Bradford, due to international research highlighting these issues via parental interviews. These findings therefore may reflect challenges both nationally and internationally within DCD pathways. There is an urgent need for: (i) clear communication across different services (with consistency in terminology), and (ii) a more collaborative and integrated approach to assessment, diagnosis, and support in order to help these children thrive.

## Introduction

Historically, children with poor sensorimotor skills have been given a range of diagnostic labels including 'Clumsy Child Syndrome' and 'Dyspraxia'. However, the research community investigating childhood onset sensorimotor skill deficits has converged on the use of the diagnostic construct of 'Developmental Coordination Disorder (DCD)' as described within the Diagnostic and Statistical Manual 5th edition (DSM-V) [1]. DCD is characterised by difficulties with fine and gross sensorimotor skills that "present in early childhood" and "significantly and persistently" impact on activities of daily living (ADLs) and/or education [1]. Global estimates suggest DCD has a prevalence of around 5–6% in the population of school-aged children, [2] making it one of the most common neurodevelopmental disorders [3]. It is important to note that the standardised use of DCD within the international research community does not imply that DCD is a standard international diagnosis. In Europe, clinicians also use the International Classification of Diseases 10th edition (ICD-10; [3]), usually because it holds legal status in their country, whereas DSM-V does not. Within the ICD-10 'Specific Developmental Disorder of Motor Function' (SDDMF) is the equivalent diagnosis given [4] although the implementation of ICD-11 will see a change in terminology to 'Developmental Motor Coordination Disorder' [5].

The international consensus to use DCD as a standardised category for research has created a rich knowledge base about sensorimotor skill difficulties. Challenges with these skills have been found to be a major risk factor for poor educational attainment [6,7], physical health [8–10], mental health [9,10], and social development [11,12]. It is crucial, therefore, that children with sensorimotor difficulties are identified

and supported in a timely manner to ensure they can have the opportunity to thrive in all aspects of their lives. This is reliant, however, on effective assessment and support pathways. Unfortunately, despite extensive efforts from research and clinical communities, DCD remains largely under-recognised and under-supported in practise [13,14] with research highlighting major under-diagnosis of DCD [15]. It has been shown that extremely low numbers of children receive a diagnosis of DCD despite clinical diagnosis being the current pathway to support in the UK [15]. It is therefore perhaps unsurprising that existing research has shown that there is considerable international parental dissatisfaction with the DCD pathway, with parents reporting high levels of stress and limited knowledge amongst communities and professionals [16–18].

This parental dissatisfaction may in part be as a result of children's services (at least in the UK) having become fragmented, with single clinical specialities sometimes struggling to address complex multifactorial issues that do not clearly fall within the remit of a single service [15,19]. This is particularly evident in the case of DCD, with sensorimotor difficulties falling between the dominions of child psychiatry, paediatrics, physiotherapy, and occupational therapy in the health service alone, and this is without considering the major role sensorimotor skills play in an education context. In contrast, other neurodevelopmental disorders – such as autism – fall more neatly within a specific service (e.g., paediatrics or Child and Adolescent Mental Health Services in the case of autism). It is possible that these issues are contributing to the under-diagnosis of DCD. Nevertheless, there has been limited insights from stakeholders involved in assessing, diagnosing, and supporting children with sensorimotor difficulties, beyond a single study which looked at knowledge and awareness via a questionnaire [20]. The current study therefore aimed to understand the barriers and facilitators in the DCD pathway within a large and diverse district in the UK (Bradford) to obtain a deep understanding of the issues across the whole system.

## Methods and materials

### Design

The current research utilised a qualitative design, which allowed an exploration of lived experience via stakeholder interviews. This enabled an analysis, which presents a holistic picture of the challenges faced within the current pathway. Multidimensional triangulation was used due to bringing together multiple stakeholders, research expertise and analyses, as well as the use of the Capability, Opportunity, Motivation model of Behaviour Change (COM-B; [21]) driving methodology. Interviews focussed on one district (Bradford, UK) were completed to obtain a deep understanding of the issues from the perspectives of all different stakeholders within the system, including occupational therapy, physiotherapy, paediatrics, school nursing, special educational needs coordinators (SENDCos), physical needs teams, general practitioners (GPs; family doctors), educational psychology and individuals involved in developing and approving Education Health and Care Plans (EHCPs).

### Setting

Bradford is the fifth largest metropolitan district in England [22] with a young population – 27.9% under the age of 20 years [22]. It is ethnically diverse including South Asian, White British, Central and Eastern European communities [23]. The district encompasses some of the most deprived areas in the UK, with nearly a quarter of children living in poverty [24]. Bradford is therefore an ideal location to explore challenges within the DCD pathway as research has found that mothers living in more deprived neighbourhoods are likely to encounter more challenges in accessing clinical services [25]. In addition, families from lower socioeconomic status backgrounds generally have fewer resources to counteract the long-term negative consequences of neurodevelopmental disorders [26,27] like DCD. Thus, Bradford provides an opportunity to study clinical pathways where diagnosis may be most valuable, but also the least accessible.

The clinical pathway for children experiencing sensorimotor issues in the district typically involves GP referral to a paediatrician, who then refers to occupational therapy and/or physiotherapy for an assessment, to aid with diagnostic decisions ultimately made by paediatricians. Alternative routes of referral via education settings are also possible, but less

commonly utilised. Although clinical pathways for DCD differ by region in the UK, the key stakeholders generally remain the same, irrespective of the route to diagnosis.

## Participants

Stakeholders involved in assessing, diagnosing, and/or supporting children on the DCD diagnosis pathway were approached to take part in a 20–30 minute interview exploring barriers and facilitators related to their role. Due to the nature of Bradford being a 'City of Research' ([28], healthcare and education professionals readily collaborate with research. Stakeholders were therefore recruited through pre-existing working relationships with Born in Bradford. Of 23 individuals approached, 18 participants consented to take part from across the pathway comprising of a GP, Paediatricians ($n=3$), Physiotherapists ($n=2$), Occupational Therapists ($n=2$), Physical Needs Team (specialist teachers who provide support for children with physical needs; $n=3$), a Special Educational Needs and Disabilities Coordinator (SENDCo), School Nurses ($n=4$), an Educational Psychologist who has a role on the Education Health and Care Plan (EHCP) panel, and a healthcare professional involved in EHCP panels (as EHCPs determine whether the school receives additional funding to support the needs of a child with Special Educational Needs and Disabilities [SEND]). Recruitment for this study started on 6th December 2023 and ended on 2nd August 2024.

## Materials

Semi-structured interview questions were generated to investigate barriers and facilitators in the DCD pathway. Questions were tailored based on each stakeholder's role, however, where questions were relevant for all stakeholders such as those surrounding knowledge, resource constraints, and beliefs surrounding roles and responsibilities, these were kept consistent across interviews. Through previous interactions with stakeholders involved in the pathway via associated Born in Bradford projects, it was evident that there was a need to better understand insider perspectives from those involved with assessing and supporting children with DCD. Particularly given that previous research has focused on the lived experience of the families [16], rather than those delivering assessment and support. Questions were therefore generated based on the literature to unpick the experiences of families, via the voices of key stakeholders (for a full interview agenda see S1 Appendix).

## Procedure

Prospective participants were sent information sheets. If participants were happy to take part in the interviews they filled in written consent forms prior to the interview. Individual interviews were conducted either in the stakeholder's place of work or online (via Microsoft Teams) according to participant preference. All interviews were recorded to facilitate transcription. Interview data were transcribed by the research team, during which time confidential information was removed, and all data were anonymised. Member checking was implemented to ensure transcripts were reflective of the participants' true feelings and improve the credibility and trustworthiness of the data [30]. Participants had two weeks to review and amend transcripts, before the data were analysed. Ethical approval for this study was granted by the University of Bradford Ethics Committee (reference: E1026).

## Analysis

The COM-B model [21] was used within a theoretical thematic analysis to deductively analyse the data. The COM-B model [21] proposes that capability, opportunity, and motivation impact upon the likelihood of a behaviour occurring (in this case, assessing and supporting children with sensorimotor skill difficulties). The COM-B model underpins the Behaviour Change Wheel [29], which was ideal for this study as it enables researchers to understand barriers to behaviours, and subsequently match evidence-based behaviour change techniques/ interventions to these barriers to effectively facilitate

the behaviour of interest. Having researched thematic analyses, the decision was made to align the current theoretical analysis to Braun and Clarke's (2006) version [31]. The six stages of analysis as cited by Braun and Clarke (2006) were clear and concise, which served the need for transparent clarity.

The six stages are: (i) Familiarising yourself with the data: Data were transcribed, each transcription was read and re-read, which allowed initial ideas to be noted (based upon the COM-B model); (ii) Generating initial codes: The research team systematically coded any interesting features within and across the data set, collating data relevant to each COM-B code; (iii) Searching for themes: Codes were collated into the COM-B themes – all relevant data was subsequently gathered to substantiate the COM-B themes; (iv) Reviewing themes: The tapestry of the COM-B themes were reviewed to confirm appropriateness of the coded extracts, and substantiation of the entire data set more broadly; (v) Defining and naming themes: Analysis continued to refine the specifics of each theme in accordance with the COM-B model; (vi) Producing the report: The research team drew upon the most vibrant and compelling data extracts, which here were reaffirmed as relational to the research aims and literature underpinning the current report.

A minimum of 3 members of the authorship team analysed each transcript, which allowed disagreements and contrasting interpretations to be discussed and considered until a consensus was reached.

## Results

The following themes illuminated several key barriers related to capability, opportunity, and motivation in assessing and supporting children with sensorimotor skill difficulties in Bradford.

### Capability

Within the COM-B model, capability refers to both psychological (e.g., knowledge, decision processes) and physical (e.g., skills) abilities. [21].

It became apparent throughout the interviews that there was a lack of consistency in terminology related to the diagnosis of clinical sensorimotor skill difficulties. Historically, childhood onset sensorimotor difficulties were diagnosed as dyspraxia, however in recent years, this diagnostic label shifted to DCD (DSM-V; [1]) and SDDMF (ICD-10; [4]). In Bradford (and some NHS trusts across the UK), diagnoses are aligned to the ICD-10 framework. This terminology will shift again in ICD-11 (due to become statutory in the UK by 2026) to Developmental Motor Coordination Disorder [5]. It became clear that dyspraxia was still the most used term by clinicians and education professionals when talking to families, despite this not being a formal diagnosis for early onset sensorimotor difficulties (and thus does not grant access to resources and support). This led to confusion at times:

*'People don't realise that dyspraxia and DCD are the same thing. So, when we say DCD, they're thinking, oh, they didn't diagnose dyspraxia… so I have to be very specific when I talk to parents'* [Participant 6; Paediatrician]

*'So if I diagnose a child I put the word DCD, I don't use any other terms, but I get the reference from primary care saying dyspraxia and even from school.'* [Participant 5, Paediatrician]

*'DCD is a health diagnosis, fine motor and gross motor…dyspraxia has a processing element as well. So is the education side of things too.'* [Participant 9, Physical Needs Team]

*'The terminology I think, I think it changed. It used to be known as DCD. I can't even think which way round it is anymore, but then they brought it in. So, I think it used to be known as dyspraxia and now I think they call it developmental coordination disorder.'* [Participant 1, Occupational Therapist]

No participants interviewed recognised or used the term Specific Developmental Disorder of Motor Function, despite the ICD-10 framework being used for diagnoses in the NHS:

*'Never come across it [Specific Developmental Disorder of Motor Function].' [Participant 3, Occupational Therapist]*

*'No I haven't heard of it [Specific Developmental Disorder of Motor Function] but it sounds like they mean developmental delay that is caused by reduced motor skills? [Participant 4, Physiotherapist]*

Across roles within the pathway, many participants were unable to give a comprehensive overview of the challenges associated with DCD, diagnostic criteria, and the pathway for assessment and support in the district:

*'When you say coordination concerns is that, for example, uh difficulties in walking or falling over?' [Participant 13, General Practitioner]*

*'In all honesty, before we [as a team, in preparation for this interview] chatted a week ago, I didn't know, I couldn't have given you that answer.' [Participant 9; Physical Needs Team]*

*'Maybe six weeks ago somebody had posted something on LinkedIn [about DCD], I actually took a picture of that... [it was] quite an eye opener... I have expanded my history [questions for parents] a bit more since… I'm not even sure what exactly the Bradford DCD pathway is…that's another reason I struggle…when do I decide to refer children to the OT? Because I don't want to waste their time' [Participant 6, Paediatrician]*

This confusion around terminology and diagnostic criteria were ultimately highlighted in the reality of high levels of inappropriate referrals into clinical services:

*'At a guess, I would say a ¼ to ⅓ [of the children we see actually have motor skill difficulties]...' [Participant 4, Physiotherapist]*

When asked for common reasons underpinning inappropriate referrals, co-occurring and conflicting diagnoses, as well as general population activity levels were highlighted as relevant:

*'Inactivity. So, standard questions, probably, would, by the paediatrician would be 'Can you swim?', 'Can you ride a bike?'. And no, they can't, but it's because they've never tried. So I'd say it's inactivity. Just lack of experience of skills.' [Participant 2, Physiotherapist]*

*'So I come across some children where… they have just been put inside their pram or, you know, in the pushchair most of the day. So, obviously if they are delayed, we cannot call that as a, you know, like coordination difficulties because they haven't had the opportunity to develop on the skills.' [Participant 5, Paediatrician]*

*'So, usually comes as a surprise to parents but that then takes us down a different line because that child then hasn't got fine motor difficulties, but they have got difficulties with fine motor tasks because this child is hypermobile.' [Participant 3, Occupational Therapist]*

*'Autism or ADHD I know you can have a dual diagnosis with autism but when autism is the more prevailing factor then it isn't always DCD as it is the learning that is the cause of the coordination problems, i.e., their motor skills are in line with their learning…' [Participant 4, Physiotherapist]*

## Opportunity

Within the COM-B model, opportunity refers to physical (e.g., environment, resources) and social (e.g., culture, values) factors that influence behaviour [21].

Throughout the DCD pathway, it was highly evident that resource constraints were impacting on support available for children with sensorimotor difficulties. For example, low levels of staffing impacting time available for these children was revealed:

*'Staffing is the biggest one [barriers to supporting], because we just can't see these children in a timely manner. Because we only have… 2 and a half OTs, and that's for the whole of the neurodisability service. So, if you compare us to, um, the rest of the ICS [Integrated Care System], um, I mean it's just shocking…...previously, when we were better staffed, we used to run [intervention] groups, lots of groups, we did all kinds of groups. We don't run any of those anymore because we don't have the staffing to do it. So, it's all sign-posting to things available in the community…'* [Participant 2, Physiotherapist]

*'There's never enough time. There's never enough staff. There's always too many kids'* [Participant 3, Occupational Therapist]

*'Always staffing, there's shortage of staffing everywhere, and also, I think that appointment attendance rate…If the child didn't attend, they get discharged and we end up seeing the child in 6-8 months, back to square one because they hadn't had the assessment'* [Participant 7, Paediatrician]

*'I think generally that is always the thing that, that staff, that school struggle with is time and staff, isn't it? And being able to free people up to do it, yeah.'* [Participant 11, SENDCo]

These staffing challenges were compounded by DCD being perceived as low priority, and long waiting lists for assessment and support for children with sensorimotor skill difficulties:

*'DCD will always be lower priority to the ones with a neurodisability [such as cerebral palsy, and complex genetic conditions]'* [Participant 2, Physiotherapist]

*'If Autism is the main diagnosis, then over time the DCD might just trickle off and disappear [from their medical records].'* [Participant 6, Paediatrician]

*'Our service covers the whole of neurodisability, [developmental] coordination disorder is low on the priority list, as opposed to neurodisability, so they could be waiting a long time, maybe 9 months more, for an initial assessment.'* [Participant 2, Physiotherapist]

*'[In Bradford] we are a traded service and so schools buy blocks of time from us, maybe the equivalent of sort of between 3 and 6 assessments for the whole year. And so you would have to be in the top six of children that were causing concern to that school in order to be seen by an Educational Psychologist…I'd hazard a guess that that's not always the children who've got motor difficulties, because they're not necessarily the ones that are causing anxiety or stress to the teachers or causing disruption'* [Participant 18, Educational Psychologist].

It was also highlighted that assessment tools available were not always suitable in terms of the children being seen by services, and the resources available to clinicians:

*'It depends on the child, and it depends on their learning ability as well. Some children just... can't concentrate for that long... or they couldn't do it [Movement Assessment Battery for Children] standardised, so we would just have to measure it on a child-per-child basis.'* [Participant 1, Occupational Therapist]

*'But, again, if I see, in clinic, that a child is struggling to retain the information [during a Movement Assessment Battery for Children], um, struggling to process, I'll always make that comment. You know, 'scored very low, however*

*showed this skill, had good grasp, had good release, understood the concept', you know?'[Participant 3, Occupational Therapist]*

*'We have got murals on the wall… that's a double-edged sword because some of the kids are completely overwhelmed visually by that... No chance of them concentrating. You're asking them to throw a ball at a target at a wall and that target is resting on a mermaid's shoulder…' [Participant 3, Occupational Therapist]*

It was evident that there was a disconnect between health and education services, demonstrated by conflicting views on the role of the school in terms of their responsibility for supporting sensorimotor skill difficulties, and their role in the wider pathway:

*'Parents are being told, and are in this situation because of schools, that we can't do anything for your child until they have a diagnosis.' [Participant 6, Paediatrician]*

*'I think unless it's impacting on the children's learning and quite significantly, it won't ever get brought to my attention… staff might not be trained necessarily to pick up on things and I might not know to look out for those difficulties with coordination and motor skills.' [Participant 11, SENDCo]*

This was further evidenced when interviewing individuals involved in EHCP decisions which determine whether schools receive additional funding to support the needs of a child with Special Educational Needs and Disabilities (SEND):

*'The bit that people always forget is that schools already have a significant amount of funding to meet the needs of their children …What have you done with the £6000 [SEND budget] that you've already got? How have you supported that young person?' [Participant 18, Educational Psychologist]*

Culture was illuminated as a key source of conflict within the DCD pathway in Bradford in terms of opportunities for children to develop these skills as well as expectations around healthcare services:

*'I get quite surprised when I hear from white British families when they tell me their children are wiping their bottoms at four and five…because Asian families would tend to do it a bit longer for their kid, including feeding them and stuff like that.' [Participant 6, Paediatrician]*

*'[In areas with high South Asian and Eastern European populations] the pressure to refer to a consultant to our secondary care is high as compared to…middle class white Caucasian area… if people who are new to the UK and UK health systems, they just can't get their head around why they can't see a paediatrician straight away… [they] do not really regard or respect…a person who is not a GP' [Participant 13, General Practitioner]*

Once a child has been highlighted as potentially having a sensorimotor skill difficulty, there were discussions around whether a diagnosis and formal support structures such as EHCPs would be beneficial. This was reflective of contrasting ideologies at play within diagnostic considerations:

*'I'm not sure what the benefit of a further referral [into healthcare for diagnosis] would have been. What would they have done differently? They would have told the teacher to do the same things…if you're thinking neurodiversity would it have been useful for that person? [It] is this perennial thing about whether you have a diagnosis and whether a diagnosis helps you as an individual.' [Participant 18, Educational Psychologist]*

*'I always ask the question to parents: 'Are you seeking a diagnosis? Do you want this for your child?' Sometimes I think parents think you've got to have it, and sometimes it's just nice just to… get them to stop and have a little think,*

*because this is gonna lead to a diagnosis for your child that isn't gonna be sponged out, you can't erase it once we've done it… So, I always give them the veto option… I can see why [parents] do it, because [they] are so fraught'* [Participant 3, Occupational Therapist]

*'It's absolutely not the case [that you need a diagnosis to get an EHCP]... It's how they present in school and how they can be helped … it isn't that an EHCP solves everything. I mean, other options might be more appropriate.'* [Participant 16, EHCP Panel Member]

*'We try and turn it around a little bit, in terms of, rather than looking at getting a diagnosis, which I know a lot of families hold a lot of weight with, let's unpack it a little bit and let's find out what the issues are around how… How is it impacting you?'* [Participant 8, Physical Needs Team]

### Motivation

Within the COM-B model, motivation refers to automatic (e.g., emotion) and reflective (e.g., professional role and identity) thought processes that influence behaviour [21].

There was a general sense of frustration surrounding overlapping services and a lack of clear understanding about roles within the DCD pathway, leading to confusion and conflict around responsibilities:

*'Referrals might just come into physiotherapy, might just come into OT, or might come in as a joint referral, or 2 separate referrals for the same child'* [Participant 2, Physiotherapist]

*'Or it may be that a child is referred to us to help with a diagnosis, so we can't diagnose as clinicians, we have to present evidence to a consultant who would then, erm, make that diagnosis.'* [Participant 3, Occupational Therapist]

*'I think it's because our pathway…is not very clear and it again varies from different areas even in the same place [between the two trusts within the Bradford District]. [Participant 5, Paediatrician]*

*'So we do liaise with colleagues within health, so physio and OT if they're involved, but clearly, we don't want to replicate work that's being done by them… sometimes we do have overlap…'* [Participant 8, Physical Needs Team]

Across the DCD pathway, many participants were highly conflicted about the level of support available for children with sensorimotor skill difficulties, particularly those that might not meet the threshold for diagnosis, but would still benefit from additional support:

*'They will still have struggles; they still have difficulties. Just because they don't fall below that [bottom 5th percentile threshold], doesn't mean they don't need any help.'* [Participant 1, Occupational Therapist]

*'How can we make sure the right children get the right level of support at the right time?'* [Participant 8, Physical Needs Team]

*'I understand schools have got less support staff because they've got less money. And so because of get less money, children with special needs are more expensive because they need support time…quite often in referrals for EHCPs they [the school] say we cannot afford to keep supporting this child, which is diabolical... The system is a bit broken..."* [Participant 18, Educational Psychologist]

### Discussion

This study aimed to understand the pathway for the assessment, diagnosis, and support of DCD in a large and diverse UK district, and the associated barriers and facilitators. Despite the research team setting out to identify facilitators in the

pathway, no key themes emerged around this across the COM-B framework. The findings painted a clear picture of a system that does not sufficiently cater for the needs of children with sensorimotor difficulties. This was particularly surprising given the demographics of the city of Bradford and its high levels of deprivation [24] with relatively high proportions of families from ethnic minority groups [23]. It is well established that these demographic characteristics are associated with a higher prevalence of sensorimotor difficulties [32–34]. Given the district's demographic profile, it might be expected, therefore, that the services in Bradford would be geared towards supporting the needs of children with sensorimotor difficulties. There was a clear consensus across all participants, however, that current pathways were not fit for purpose. Demographic and cultural factors were highlighted specifically within interviews, not just around children's sensorimotor ability (specifically regarding differences in activities of daily living), but also around parental expectations of care. In addition, interviews illuminated a large disconnect between health and education regarding sensorimotor difficulties, with roles and responsibilities blurred. This lack of synergy across services and a lack of joined up access between health and education appears to result in children being missed. This has been highlighted as a problem nationally [35].

Interviews highlighted that many health service referrals for the assessment of DCD are inappropriate – in part because difficulties could be clearly attributed to a lack of opportunity related to skill acquisition. This phenomenon is likely to have been exacerbated by the COVID-19 pandemic with reports of lockdowns having a negative impact on children's physical activity levels [36,37] and, ultimately, their opportunities to explore the world. This could in part be due to a lack of consistent and comprehensive knowledge about the diagnostic criteria for DCD and methods to support sensorimotor skill development more broadly, which was demonstrated across stakeholders. One example was Occupational Therapy highlighting high proportions of children referred for DCD who turned out to have hypermobility. This lack of knowledge was also evident in education settings, with a SENDCo noting that teachers do not receive training on DCD or sensorimotor skill development. This is particularly worrying given sensorimotor skills feature heavily in the Early Years Curriculum (ages 3–5 years old) [38] and the Key Stage 1 Physical Education (ages 5–7 years old) [39], and form a core component of how children's academic progress is assessed (e.g., handwriting). This lack of knowledge is not isolated to Bradford as previous research has shown the same problems internationally [40].

This lack of knowledge could, in part, be attributed to the array of terminology that is being used within research and clinical fields. In this study alone, we identified five different terms being actively used to describe DCD. Previous research has also highlighted 10 diagnosis or 'finding' codes within healthcare to depict childhood onset sensorimotor difficulties [15]. Discussion with stakeholders highlighted that the inconsistency between terminology was often a source of confusion and tension around information sharing. Schools and parents appeared to be more familiar with the term dyspraxia, which is not a distinct diagnosis in any medical classification system for childhood onset sensorimotor difficulties [41]. This is at odds with medical professionals that often reported using the the term DCD. It was notable that the relevant diagnostic classification for the district and NHS in the UK (SDDMF; ICD-10 categorisation; [4]) was not known by anyone involved in the relevant health service pathways. In addition, in academic literature, there is also a multitude of terminology used when referring to sensorimotor development, including motor skills, motor competence and movement skills, which are often used interchangeably [42,43]. These terms, however, refer to infinite combinations of skills including fine and gross motor and activities of daily living (which are complex multifactorial categories in isolation). It is therefore important to increase specificity within research and healthcare.

This lack of consistency in terminology nationally, and internationally [44], is likely contributing to the systemic lack of awareness, assessment, and support for children with DCD. A shift towards universal terminology, accepted and understood by healthcare, education and wider support systems is paramount. This could play a pivotal role in improving identification, assessment and support for children that are often overlooked. As key themes arose around Capability Psychological (knowledge), Capability Physical (inappropriate referrals), and Opportunity Social (culture), the Behaviour Change Wheel [29] suggests implementing education interventions would be beneficial. There is therefore a need to develop evidence-based training for health and education professionals, communities and families to ensure consistent and thorough understanding of sensorimotor development as well as how to identify and support children with sensorimotor difficulties.

Assessment and support were also found to be heavily influenced by a severe lack of resources, unfortunately culminating in DCD being considered low priority (by the system, rather than those working within it). Staffing barriers were seen across the board with all stakeholders highlighting that the lack of resources often resulted in minimal time available for children with DCD, which is reflective of national struggles within the NHS [45] This was evidenced by services no longer being able to offer interventions, as well as many schools applying for Education Health and Care Plans (EHCP) for children with sensorimotor difficulties. However, these EHCP requests are unlikely to be successful as: (i) these children are not likely to receive assessment from an Educational Psychologist due to them causing less 'disruption' than children with other needs; and (ii) expectations that these children can be cared for within existing SEND provision in schools. As the process for EHCPs varies by region in the UK [46], children with DCD— and SEND more broadly— are subject to a postcode lottery in terms of available support. A culmination of these challenges ultimately results in DCD being a low priority for both healthcare and education, compared to other neurodevelopmental disorders (e.g., autism and ADHD).

Finally, there was concern about the diagnosis-led approach to supporting children with sensorimotor difficulties. Specifically, stakeholders were concerned that: (i) children are being missed by the current system; and (ii) children who are not eligible for a diagnosis, but would benefit from additional support, are not getting the help they require. Participants made it clear that we need to move towards a more integrated system, where health and education work synchronously to provide early and consistent support for sensorimotor skill difficulties, without having to rely on a diagnosis to trigger support systems. This would mitigate confusion around roles and responsibilities and ensure consistent terminology and advice for families. Moving towards a needs-led system may also help reduce clinical waiting lists for diagnosis, as fewer children who have lacked the opportunity to develop sensorimotor skills will be referred to specialist services. This in turn has the potential to be beneficial for the many individuals for whom diagnosis plays a pivotal role in quality of life, and a sense of self in the lived experience of DCD [47,48]. Such approaches would also align with the Behaviour Change Wheel [29] suggestions for barriers in Opportunity Physical (time and resources) and Automatic Motivation (concerns re: diagnostic threshold) – advocating for restructuring the environment.

## Future directions

One way a more integrated system could be achieved is by incorporating a universal offer within schools and/or communities. Recent research has shown promising signs of low-cost universal approaches in schools being an effective first step in the early identification of sensorimotor difficulties [49,50]. In addition, evidence suggests that school-based sensorimotor interventions are effective at improving these skills and thus, education settings could also play a vital role in supporting children that would benefit from additional support [51]. This would align with education curricula as sensorimotor skills are heavily embedded in the Early Years [38], play a key role in early Physical Education [39] and fine motor skills such as handwriting form vehicles of assessment throughout school life. Universal approaches could play a vital role in identifying children that may be missed, and therefore help to reduce health inequalities, and facilitate access to targeted support for ethnic minorities and those living in more deprived areas. It would also facilitate a more integrated system where educational establishments are key stakeholders in the provision of care for children and young people. Importantly, however, these new pathways must be co-produced with individuals with lived experience to ensure they benefit users. Importantly, challenges associated with DCD persist into adulthood [52], and for those who miss out on diagnosis in childhood, there needs to be a viable pathway for adult diagnosis [53]. However, currently, there is a lack of a standardised pathway for diagnosing beyond childhood [54] which for many necessitates the use of private healthcare [46,47].

## Limitations

One limitation of this research is that two of the authors had pre-existing working relationships with two of the participants. However, the two authors were not involved in interviews with these participants which mitigated for any potential bias. In addition, the participants were largely healthcare professionals and previous teachers (physical needs team). Within healthcare professions interviewed, the small sample size for each role reflects the nature of small teams with

a responsibility for DCD in the Bradford district. As recruitment was limited to professionals that have previous working relationships with Born in Bradford, this limited the scope of potential participants. For example, only one GP was interviewed, and thus it is difficult to generalise the quotes from this individual to all GPs in the area. However, this GP was well connected to other GPs in Bradford via a reducing inequalities district wide initiative, so had awareness of current issues. Similarly, only one of the participants was currently teaching (SENDCo). Even though effort was made to recruit more current school staff, a staffing crisis in Bradford hindered their capacity to engage with research [55,56]. Within the UK, there is no standardised pathway for assessment, diagnosis and support as each area is given the autonomy to build systems that meet the needs of their specific population [57,58]. It is therefore possible that other areas experience different barriers and facilitators in their pathways. However, it is well-established in international literature that there is a lack of knowledge, support and resources [44]. The current findings support this reality, and therefore likely, at least in part reflect some commonality in challenges faced.

## Conclusion

These interviews found unequivocal evidence that the pathway for assessment and support of DCD is not 'fit for purpose'. The problems identified by a wide range of stakeholders across health and education suggest strongly that children with sensorimotor difficulties are being failed in Bradford, which may reflect some of the challenges faced both across the UK and internationally [16–18]. These findings show unambiguously that there is a need to reimagine how public services support these children and their families given the well-established deleterious impact of these problems if support is not provided. Health and education sectors need to collaborate more effectively, and transition towards a more integrated approach for sensorimotor development to give children the best possible opportunity to fulfil their potential. Evidence-based training needs to be co-produced with education and health professionals, communities and individuals with lived experience to raise awareness and understanding around sensorimotor development and the challenges faced by those with DCD.

## Supporting information

**S1 Appendix. Interview Agendas.**
(DOCX)

## Author contributions

**Conceptualization:** Lucy H. Eddy, Rebecca Murray.

**Data curation:** Lucy H. Eddy, Nat K. Merrick, Cara E. Staniforth, Jade L. Jukes, Rebecca Murray.

**Formal analysis:** Lucy H. Eddy, Nat K. Merrick, Cara E. Staniforth, Jade L. Jukes, Rebecca Murray.

**Funding acquisition:** Lucy H. Eddy, Mark Mon-Williams.

**Investigation:** Lucy H. Eddy.

**Methodology:** Liam J. B. Hill, Rebecca Murray.

**Supervision:** Lucy H. Eddy, Liam J. B. Hill, Rebecca Murray.

**Writing – original draft:** Lucy H. Eddy.

**Writing – review & editing:** Nat K. Merrick, Cara E. Staniforth, Jade L. Jukes, Liam J. B. Hill, Mark Mon-Williams, Farid Bardid, Rebecca Murray.

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
