## [Decision Letter · Decision Letter 0]

17 Oct 2025

Dear Dr. Lucy H. Eddy,

Thank you for submitting your manuscript to PLOS ONE. After careful consideration, we feel that it has merit but does not fully meet PLOS ONE’s publication criteria as it currently stands. Therefore, we invite you to submit a revised version of the manuscript that addresses the points raised during the review process.

We look forward to receiving your revised manuscript.

Kind regards,

Tadashi Ito

Academic Editor

PLOS ONE

Journal Requirements:

“The work of the lead author (L.H. Eddy) and co-author (C.E. Staniforth) was supported by a grant from the Waterloo Foundation (ref: 27665413). M. Mon-Williams was supported by a Fellowship from the Alan Turing Institute. The work was conducted within infrastructure provided by the Centre for Applied Education Research (funded by the Department for Education through the Bradford Opportunity Area) and ActEarly: a City Collaboratory approach to early promotion of good health and wellbeing funded by the Medical Research Council (grant reference MR/S037527/). M. Mon-Williams’ involvement was supported by the National Institute for Health Research Yorkshire and Humber ARC (reference: NIHR20016). The views expressed in this publication are those of the author(s) and not necessarily those of the National Institute for Health Research or the Departments of Health and Social Care or Education. The funders had no role in study design, data collection and analysis, decision to publish, or preparation of the manuscript.”

“The work of the lead author (L.H. Eddy) and co-author (C.E. Staniforth) was supported by a grant from the Waterloo Foundation (ref: 27665413). M. Mon-Williams was supported by a Fellowship from the Alan Turing Institute. The work was conducted within infrastructure provided by the Centre for Applied Education Research (funded by the Department for Education through the Bradford Opportunity Area) and ActEarly: a City Collaboratory approach to early promotion of good health and wellbeing funded by the Medical Research Council (grant reference MR/S037527/). M. Mon-Williams’ involvement was supported by the National Institute for Health Research Yorkshire and Humber ARC (reference: NIHR20016).”

“The work of the lead author (L.H. Eddy) and co-author (C.E. Staniforth) was supported by a grant from the Waterloo Foundation (ref: 27665413). M. Mon-Williams was supported by a Fellowship from the Alan Turing Institute. The work was conducted within infrastructure provided by the Centre for Applied Education Research (funded by the Department for Education through the Bradford Opportunity Area) and ActEarly: a City Collaboratory approach to early promotion of good health and wellbeing funded by the Medical Research Council (grant reference MR/S037527/). M. Mon-Williams’ involvement was supported by the National Institute for Health Research Yorkshire and Humber ARC (reference: NIHR20016). The views expressed in this publication are those of the author(s) and not necessarily those of the National Institute for Health Research or the Departments of Health and Social Care or Education. The funders had no role in study design, data collection and analysis, decision to publish, or preparation of the manuscript.  “

Reviewers' comments:

Reviewer's Responses to Questions

**Comments to the Author**

1. Is the manuscript technically sound, and do the data support the conclusions?

Reviewer #1: Partly

2. Has the statistical analysis been performed appropriately and rigorously?

Reviewer #1: N/A

3. Have the authors made all data underlying the findings in their manuscript fully available?

Reviewer #1: Yes

4. Is the manuscript presented in an intelligible fashion and written in standard English?

Reviewer #1: Yes

Reviewer #1: Review

The system is a bit broken…” a qualitative exploration of barriers in the pathway for

diagnosing Developmental Coordination Disorder/ Specific Developmental Disorder of

Motor Function.

Overall, this study is well-written and gives an interesting insight in the perception of professionals regarding the diagnostic pathway which may be related to the intervention opportunities for children with DCD. It has been an extensive study to conduct and process all the information. However, only a small number of participants was included which could not cover the entire district. The authors need to reconsider not to generalize the outcomes, since the level of knowledge amongst other parts of Bradford (and the rest of the UK) can be different. There are only three key-topics extracted which were defined beforehand. What topics came up through the transcription procedure? I wonder why prompts were chosen to be included? By including these you intend to steer the answers? For example budgets/resources are no prompts for PTs or OTs?

The paper can improve by making it more appropriate and into line with current international knowledge.

1. Title: The system is a bit broken… if this is one answer on one question of one of your participants, do you believe it completely covers the topic? Besides it is not only aimed at diagnosing, correct? And you were also looking for facilitators?

2. Abstract: the conclusion that “These findings are unlikely to be unique to

Bradford and appear to reflect national and international health service challenges” is presented too strongly, without evidence whatsoever. Where do you support ‘appear’ upon? And what is the connection with international situations? Please, rephrase

3. In the DSM5 the neurodevelopmental motor disorders include DCD, stereotypic movement disorder and tic disorders. The term “Specific Developmental Disorder of Motor Function” is hardly used in research and rehabilitation and it is agreed upon that DCD is supposed to be the commonly used term (as confirmed in the first paragraph of the intro). Please, remove SDDMF to increase consistency of the terms being used. You can introduce it in the Method section, since the results confirm that this term is not being used.

4. In the abstract the abbreviations (COM-B model) should be written out in full at first use.

5. Final paragraph in the introduction:”the parental dissatisfaction is consistent with the observation that….” Whose observation? And in what part of the UK was it observed?

6. “sparse insights”? How can you support this or do you have a reference?

7. Design: interviews focused on one district (Bradford). However, I don’t think you can say this since Bradford borough is an enormous district with a huge population. I don’t think interviewing 1 GP would cover the whole area. Please, rewrite and be more specific on the area or maybe the links to a hospital, rehab center or how many schools to know what this qualitative study represents within this huge area.

8. In total 18 stakeholders were recruited: how many were invited?

9. COM-B: please write out in full first time and introduce the model more clearly. Specifically, how you directed capability, as originally referred to an individual’s psychological and physical ability to participate in an activity: which questions were aimed for this process? And which external factors were asked to present the opportunity of which behavior and finally, motivation: which questions referred to the conscious and unconscious cognitive processes that direct and inspire change. Please refer to the first authors that introduced the model.

10. Behaviour Change Wheel: please, again explain or introduce earlier. We need to know how you used this model and based upon this model why you presented the answers you chose to present as results.

11. “This terminology will shift again in ICD-11 (due to become statutory in the UK by 2026) to Developmental Motor Coordination Disorder (28)”. Are you sure you can say this based on the reference of 2022? Since then, it has become common knowledge that DCD is not limited to only motor deficits, but also poor executive functioning (planning, execution, anticipatory control and working memory)?

12. Can you explain what you mean by Pre and post FUNMOVES? This needs some clarification and a reason why you included this in your questions.

13. In the discussion you focus in the second paragraph on opportunity. However, why not start with capability to keep the order consistent?

14. What is an Early Years Curriculum during Key Stage 1 Physical Education: this will probably be known in the UK, but since this journal is an international journal you better explain the content and ages involved.

15. In the third paragraph of the discussion, you only present the fact that there is not enough knowledge regarding diagnosing DCD consistently. It might be good that you at least bring forward the four diagnostic criteria for DCD according to the DSM5 to improve the consistency in diagnosing for DCD. This might be the transfer from the third to fourth paragraph. In some countries these criteria are accepted well by health and educational professionals.

16. “Participants made it clear that we need to move towards a more integrated system, where health and education work synchronously to provide early and consistent support for sensorimotor skill difficulties, without having to rely on a diagnosis to trigger support systems.” This may be one side of the coin, the other is that many children feel for the first time ‘understood’ when they understand what DCD means. Many parents wished they would have known this many years before, to prevent them from becoming angry with their ‘clumsy’ child. Knowledge for the parents and children is the most important factor that interferes with self-esteem and preventing anxiety or depression when becoming grown-ups. You may give this as a limitation, since your interviews were not directed at parents of children with DCD or children with DCD.

17. The future directions do not take into account that DCD will not be overgrown and needs understanding of the developing child and their parents and environments. Why not?

18. I am not sure where the evidence based training comes from, since this is not the scope of this paper.

19. Limitations: number of participants is small and very variable, please address.

20. These results cannot account for the whole of the district.

**Do you want your identity to be public for this peer review?** For information about this choice, including consent withdrawal, please see our Privacy Policy

Reviewer #1: **Yes:** Dorothee Jelsma

---

## [Author Response · Author response to Decision Letter 1]

21 Nov 2025

Reviewer #1:

Overall, this study is well-written and gives an interesting insight in the perception of professionals regarding the diagnostic pathway which may be related to the intervention opportunities for children with DCD. It has been an extensive study to conduct and process all the information.

We thank the reviewer for their kind words.

However, only a small number of participants was included which could not cover the entire district. The authors need to reconsider not to generalize the outcomes, since the level of knowledge amongst other parts of Bradford (and the rest of the UK) can be different.

The paper can improve by making it more appropriate and into line with current international knowledge.

7. Design: interviews focused on one district (Bradford). However, I don’t think you can say this since Bradford borough is an enormous district with a huge population. I don’t think interviewing 1 GP would cover the whole area. Please, rewrite and be more specific on the area or maybe the links to a hospital, rehab center or how many schools to know what this qualitative study represents within this huge area.

19. Limitations: number of participants is small and very variable, please address.

20. These results cannot account for the whole of the district.

Whilst we appreciate that Bradford is a large district, we cannot publish links to specific hospitals, practices, teams, or schools as this could potentially breach anonymity. Given the few professionals with a responsibility for DCD in the District, any such links could mean their responses are identifiable. For example, in the Bradford NHS Trust there are two Occupational Therapists with a responsibility for DCD, and two Physiotherapists with a responsibility for DCD, all of which were interviewed. If we were to identify the hospital where they work that would mean their responses could be easily attributed to each individual. We appreciate that only interviewing one GP is a limitation, and as such, have included this within the discussion section. We agree that the results cannot be generalised across the UK fully, due to each NHS trust using different pathways, professionals and services for care related to DCD. We have also included this in the discussion, however have also acknowledged that these challenges reflect results from international studies with parents too.

There are only three key-topics extracted which were defined beforehand. What topics came up through the transcription procedure? I wonder why prompts were chosen to be included? By including these you intend to steer the answers? For example budgets/resources are no prompts for PTs or OTs?

9. COM-B: please write out in full first time and introduce the model more clearly. Specifically, how you directed capability, as originally referred to an individual’s psychological and physical ability to participate in an activity: which questions were aimed for this process? And which external factors were asked to present the opportunity of which behavior and finally, motivation: which questions referred to the conscious and unconscious cognitive processes that direct and inspire change. Please refer to the first authors that introduced the model.

Interview questions were not designed using COM-B rather responses were analysed using this framework. Questions were designed to broadly explore barriers and facilitators specific to each role. Questions were not constrained by a theoretical model. However, in order to achieve a comprehensive analysis we used deductive codes related to the COM-B model as it is a well-established evidence-based model that allows for intervention(s) to be suggested to ameliorate any barriers identified. Therefore, the three themes presented were deductive based on the COM-B analysis of data. In terms of questions and prompts chosen, these were generated to investigate barriers and facilitators in the DCD pathway, tailored based on each stakeholder’s role. Through previous interactions with stakeholders involved in the pathway via associated Born in Bradford projects, it was evident that there was a need to better understand insider perspectives from those involved with assessing and supporting children with DCD, as previous research has focused on the lived experience of the families. Questions were therefore generated based on the literature to unpick the experiences of families, via the voices of key stakeholders. Where questions were relevant for all parties (e.g. resources, time and staffing) these were consistent across interviews (e.g. Q15 for OT and Physios). We apologise for the confusion and have rectified this within the manuscript.

1. Title: The system is a bit broken… if this is one answer on one question of one of your participants, do you believe it completely covers the topic? Besides it is not only aimed at diagnosing, correct? And you were also looking for facilitators?

Whilst we were also looking for facilitators within pathway, unfortunately none were illuminated within the lived experience of clinicians and professionals involved in the pathway. We believe that the title ‘The system is a bit broken’ accurately reflects all of the issues presented in the paper – both within clinical services (including diagnosis) and beyond. For example, it perfectly highlights issues with resources, waiting lists, conflicting terminology, and a lack of integration between health and education. Moreover, although the focus was on the diagnostic pathway, this is the gatekeeper to support, and thus this remains relevant for support too.

2. Abstract: the conclusion that “These findings are unlikely to be unique to

Bradford and appear to reflect national and international health service challenges” is presented too strongly, without evidence whatsoever. Where do you support ‘appear’ upon? And what is the connection with international situations? Please, rephrase.

We have rephrased this to: These findings are unlikely to be unique to Bradford, due to international research highlighting these issues via parental interviews. These findings therefore may reflect challenges both nationally and internationally within DCD pathways. Where this argument is presented in the discussion, such parental literature has been cited to justify.

3. In the DSM5 the neurodevelopmental motor disorders include DCD, stereotypic movement disorder and tic disorders. The term “Specific Developmental Disorder of Motor Function” is hardly used in research and rehabilitation and it is agreed upon that DCD is supposed to be the commonly used term (as confirmed in the first paragraph of the intro). Please, remove SDDMF to increase consistency of the terms being used. You can introduce it in the Method section, since the results confirm that this term is not being used.

Whilst we agree that DCD is the more commonly used term, we need to include context about this in the introduction due to the questions asked. Specifically, because although clinicians use the terminology DCD in the UK, the actual diagnosis children receive is Specific Developmental Disorder of Motor Function (SDDMF) as legally the UK use the ICD framework for diagnosis. We have removed all combined mention of DCD/SDDMF throughout the manuscript after this, to improve for consistency and ease of understanding. This aligns with the EACD report which highlights that although DCD is a well recognised term, in several European countries, ICD1- has legal status, and thus the term SDDMF has to be used for these countries (Blank et al., 2019).

4. In the abstract the abbreviations (COM-B model) should be written out in full at first use.

This has been written in full in the abstract, and once in the introduction to avoid confusion.

5. Final paragraph in the introduction: ”the parental dissatisfaction is consistent with the observation that….” Whose observation? And in what part of the UK was it observed?

We have clarified this sentence in the manuscript, and added references to substantiate the point.

6. “sparse insights”? How can you support this or do you have a reference?

We have clarified this within the introduction (i.e. that a lot of the current literature looks at parental experiences, rather than those responsible on the DCD pathway). We have, however, included a reference for a study which surveyed clinicians.

8. In total 18 stakeholders were recruited: how many were invited?

We have included that 23 were approached and of those 18 consented to take part.

10. Behaviour Change Wheel: please, again explain or introduce earlier. We need to know how you used this model and based upon this model why you presented the answers you chose to present as results.

Results were presented through the lens of COM-B, not the Behaviour Change Wheel. The Behaviour Change Wheel is an associated resource, which can be used to identify potential interventions to associated barriers and facilitators found within the COM-B framework. We have clarified this in the methods (analysis) section.

11. “This terminology will shift again in ICD-11 (due to become statutory in the UK by 2026) to Developmental Motor Coordination Disorder (28)”. Are you sure you can say this based on the reference of 2022? Since then, it has become common knowledge that DCD is not limited to only motor deficits, but also poor executive functioning (planning, execution, anticipatory control and working memory)?

The reference relates to the new ICD-11 framework, which is due to be implemented by law in 2026 (new reference included to justify that this is becoming statutory). Despite the knowledge from literature that DCD encompasses more than just challenges with motor skills, unfortunately this is the new terminology being introduced by the World Health Organisation in their updated International Classification of Diseases. Whilst we agree this is problematic, it is beyond the scope of this paper to discuss this.

12. Can you explain what you mean by Pre and post FUNMOVES? This needs some clarification and a reason why you included this in your questions.

The year before the interviews took place, the OT and Physio services in Bradford adopted a universal screening and intervention tool into their practices (FUNMOVES). As this constitutes resources (both for clinicians and families), the team thought it was important to explore both pre and post the adoption of FUNMOVES due to the recency of this addition. Clarification around this has been added to the Supplementary Material 1.

13. In the discussion you focus in the second paragraph on opportunity. However, why not start with capability to keep the order consistent?

Whilst the word opportunity is mentioned in this paragraph, it is in relation to what children are exposed to in order to develop these skills. This is linked to inappropriate referrals (i.e. a lack of parent and family doctor / paediatrician knowledge). We have clarified this in the manuscript.

14. What is an Early Years Curriculum during Key Stage 1 Physical Education: this will probably be known in the UK, but since this journal is an international journal you better explain the content and ages involved.

We have clarified the age ranges for each of these stages in education to ensure it can be understood by an international audience.

15. In the third paragraph of the discussion, you only present the fact that there is not enough knowledge regarding diagnosing DCD consistently. It might be good that you at least bring forward the four diagnostic criteria for DCD according to the DSM5 to improve the consistency in diagnosing for DCD. This might be the transfer from the third to fourth paragraph. In some countries these criteria are accepted well by health and educational professionals.

Whilst this is an interesting suggestion, there are key differences between the two diagnostic criteria (i.e. DCD vs SDDMF). To be specific, the ICD-10 does not include a requirement for ‘early onset’ of challenges, nor does it include a specification that these challenges need to ‘significantly and persistently impact on daily living’. To include one set of criteria (that is not officially used in the UK) and not present SDDMF would be problematic for this reason. Whilst we agree these differences are problematic generally, it is beyond the scope of this paper to discuss this.

16. “Participants made it clear that we need to move towards a more integrated system, where health and education work synchronously to provide early and consistent support for sensorimotor skill difficulties, without having to rely on a diagnosis to trigger support systems.” This may be one side of the coin, the other is that many children feel for the first time ‘understood’ when they understand what DCD means. Many parents wished they would have known this many years before, to prevent them from becoming angry with their ‘clumsy’ child. Knowledge for the parents and children is the most important factor that interferes with self-esteem and preventing anxiety or depression when becoming grown-ups. You may give this as a limitation, since your interviews were not directed at parents of children with DCD or children with DCD.

A needs-led system does not remove the need for diagnosis, rather will help support those children that need a diagnosis receive this faster. As you allude to this can be very powerful for children and families. We have included a sentence about this after the sentence you highlighted. Whilst we appreciate that parental and child voice in such matters are important, this was beyond the scope of the research question, as previous research has been conducted on these issues (as referenced in the introduction). This study wanted to look at the gap (i.e. clinician and pathway perspectives on why these barriers exist).

17. The future directions do not take into account that DCD will not be overgrown and needs understanding of the developing child and their parents and environments. Why not?

We were not entirely sure what you were referring to when you refer to DCD being overgrown. We interpreted this as ‘persists into adulthood’ and have sought to address this. We agree that literature shows that DCD persists into adulthood. We have included some discussion around this in the future directions section.

18. I am not sure where the evidence based training comes from, since this is not the scope of this paper.

Whilst this paper did not explicitly look at training, there was a consistently poor knowledge of DCD across sectors. The COM-B model and Behaviour Change Wheel show that one (evidence-based) way to improve knowledge is by embedding education interventions (i.e. training). The need for this to be evidence-based and informed by lived experience is clear due to the many misconceptions surrounding DCD. This is included in the discussion section.

---

## [Decision Letter · Decision Letter 1]

15 Feb 2026

“The system is a bit broken…” a qualitative exploration of barriers in the pathway for diagnosing Developmental Coordination Disorder

PONE-D-25-16312R1

Dear Dr. Lucy H. Eddy,

We’re pleased to inform you that your manuscript has been judged scientifically suitable for publication and will be formally accepted for publication once it meets all outstanding technical requirements.

Kind regards,

Tadashi Ito

Academic Editor

PLOS One

Additional Editor Comments (optional):

Reviewers' comments:

Reviewer's Responses to Questions

**Comments to the Author**

Reviewer #2: All comments have been addressed

2. Is the manuscript technically sound, and do the data support the conclusions?

Reviewer #2: (No Response)

3. Has the statistical analysis been performed appropriately and rigorously?

Reviewer #2: (No Response)

4. Have the authors made all data underlying the findings in their manuscript fully available?

Reviewer #2: (No Response)

5. Is the manuscript presented in an intelligible fashion and written in standard English?

Reviewer #2: (No Response)

Reviewer #2: (No Response)

**Do you want your identity to be public for this peer review?** For information about this choice, including consent withdrawal, please see our Privacy Policy

Reviewer #2: No

---

## [Editor Report · Acceptance letter]

PONE-D-25-16312R1

PLOS One

Dear Dr. Eddy,

I'm pleased to inform you that your manuscript has been deemed suitable for publication in PLOS One. Congratulations! Your manuscript is now being handed over to our production team.

Kind regards,

on behalf of

Dr. Tadashi Ito

Academic Editor

PLOS One